# Inputs for optimizing selection platform for milk production traits of dairy Sahiwal cattle

**Destaw Worku[¤]\*, Gopal Gowane, Rani Alex, Pooja Joshi, Archana Verma**

Animal Genetics and Breeding Division, National Dairy Research Institute, Karnal, Haryana, India

¤ Current address: Department of Animal Science, Salale University, Salale, Ethiopia
* destawworku@gmail.com

**Data Availability Statement:** All relevant data are within the paper and its Supporting Information files. However, raw milk production data along with data of pedigree file of Sahiwal cattle used in this study cannot be shared publicly. This is due to

## Abstract

The premises for the potential success of molecular breeding is the ability to identify major genes associated with important dairy related traits. The present study was taken up with the objectives to identify single nucleotide polymorphism (SNP) of bovine MASP2 and SIRT1 genes and its effect on estimated breeding values (EBVs) and to estimate genetic parameters for lactation milk yield (LMY), 305-day milk yield (305dMY), 305-day fat yield (305dFY), 305-day solid not fat yield (305dSNFY) and lactation length (LL) in Sahiwal dairy cattle to devise a promising improvement strategy. Genetic parameters and breeding values of milk production traits were estimated from 935 Sahiwal cattle population (1979–2019) reared at National Dairy Research Institute at Karnal, India. A total of 7 SNPs, where one SNP (g.499C>T) in exon 2 and four SNPs (g.576G>A, g.609T>C, g.684G>T and g.845A>G) in exon 3 region of MASP2 gene and 2 SNPs (g.-306T>C and g.-274G>C) in the promoter region of SIRT1 gene were identified in Sahiwal cattle population. Five of these identified SNPs were chosen for further genotyping by PCR-RFLP and association analysis. Association analysis was performed using estimated breeding values (n = 150) to test the effect of SNPs on LMY, 305dMY, 305dFY, 305dSNFY and LL. Association analysis revealed that, three SNP markers (g.499C>T, g.609T>C and g.-306T>C) were significantly associated with all milk yield traits. The estimates for heritability using repeatability model for LMY, 305dMY, 305dFY, 305dSNFY and LL were low, however the corresponding estimates from first parity were 0.20±0.08, 0.17±0.08, 0.13±0.09, 0.13±0.09 and 0.24, respectively. The repeatability estimates were moderate to high indicating consistency of performance over the parities and hence reliability of first lactation traits. Genetic correlations among the traits of first parity were high (0.55 to 0.99). From the results we could conclude that optimum strategy to improve the Sahiwal cattle further would be selecting the animals based on their first lactation 305dMY. Option top include the significant SNP in selection criteria can be explored. Taken together, a 2-stage selection approach, select Sahiwal animals early for the SNP and then on the basis of first lactation 305dMY will help to save resources.

there are legal restrictions for sharing the data as the data pertains to the animal breeding experiment and not for open access sharing. This is so because it contains information regarding the selection of the animals and other important information, which is supposed to be not shared in raw form. The restrictions are imposed by the Research Advisory Council of the Institute from where the data is used for analysis (ICAR-National Dairy Research Institute, Karnal 132001 - India). Such data is only available upon request (for validation purposes, if any). In such a case, the corresponding author (Destaw Worku: email: destawworku@gmail.com) should be contacted. The non-author institutional contact for the inquiry of the data is the Director ICAR-NDRI Karnal. The contact details is: dir@ndri.res.in.

**Funding:** This research was financially supported by Animal Genetics and Breeding Division of ICAR-National Dairy Research Institute, Karnal, India. The corresponding author (Destaw Worku) received the fund. The funder had no role in study design, data collection and analysis, decision to publish, or preparation of the manuscript.

**Competing interests:** The authors have declared that no competing interests exist.

## Introduction

In dairy cattle breeding program, genetic improvement of milk production traits is the primary economic activity for profitable dairy business. However, genetic improvement relies on the amount of additive genetic variance in the population and genetic relationship of traits involved in economics of dairy farm. Therefore, estimates of genetic parameters are essential for designing optimum animal breeding program and also for the prediction of selection response. Numerous studies on heritability and genetic correlations, mainly on the basis of first lactation 305 days milk yield for Sahiwal cattle have been reported in India. Genetic parameters on the basis of multiple lactations applying repeatability animal model are more accurate [1], however, lactation milk yield (LMY), 305- days milk yield (305dMY), 305-days fat yield (305dFY), 305-days solid not fat yield (305dSNFY) and lactation length (LL) of Sahiwal cows, estimates from repeatability model are rare in literature. Sahiwal being an important dairy breed of cattle in India, needs to have these estimates for designing optimum breeding program.

There is evidence across several livestock species for genetic control of phenotypic variation of complex traits [2, 3]. Traditionally, the genetics of complex traits such as milk production traits in domestic animals has been investigated without exploring the candidate genes involved, while selection has been based on estimated breeding values obtained from phenotypic records and pedigrees, and on knowledge of the heritability of each trait [4]. For the quantitative trait loci, it is difficult to find a few single nucleotide polymorphisms (SNP) which could contribute for significant proportion of variance, however, if found they can revolutionise the selection process, time required for identification of better breeding pairs and resource expenditure. The classical forward genetics approaches which are mainly focused on a single gene effect have been successful in the identification of limited number of causal genes [5]. A few examples in livestock are the autosomal Booroola fecundity gene (FecB) in sheep [6, 7], Myostatin gene (MSTN) for double muscling in beef cattle [8]. In dairy cattle, strong functional candidate genes that affect milk production traits such as diacylglycerol acyltransferase 1 (DGAT1), growth hormone receptor (GHR) and ATP binding cassette subfamily G member 2 (ABCG2) [9–11] genes have been reported. Therefore, identification of causative mutations related to milk production traits is of paramount importance given their future utility.

In this study, we analysed polymorphisms in genes associated with milk production traits and along with estimation of genetic parameters and prediction of breeding value for milk production traits in Indian Sahiwal cattle breed. Candidate genes, MASP2 (Manan binding lectine serine associated protease 2) and SIRT1 (Silent information regulator 1) were chosen on the basis of evidence of biochemical processes related to production traits, and also history of significant association with various economic traits of cattle in the literature [12–15]. However, to date, very few studies have reported the genetic association of SNP variation in MASP2 and SIRT1 genes with milk production traits in dairy cattle. Therefore, this study aimed to identify SNPs of bovine MASP2 and SIRT1 genes in Sahiwal cattle and their possible effect on the estimated breeding values (EBVs). We also aimed to estimate the genetic parameters for LMY, 305dMY, 305dFY, 305dSNFY and LL in Sahiwal dairy cattle for assessing the breeding program.

## Materials and methods

### Experimental animals

All protocols for collection of blood samples for experimental animals and phenotypic observations were approved by the Animal Care and Use Ethics Committee at ICAR-National

Dairy Research Institute, Karnal, India. Moreover, the experiments were conducted in accordance with the guidelines of Committee for the Purpose of Control and Supervision of Experimentation in Animals (CPCSEA), Ministry of Environment, Forest and Climate Change, Government of India.

## Phenotypic data collection

The original data were collected from 935 Sahiwal cows (1979–2019) with all parities maintained in ICAR-National Dairy Research institute (ICAR-NDRI), India. All herd information regarding birth date, Sire ID, Dam ID, calving dates, drying date and lactation performance were obtained from history-cum-pedigree sheets. The traits used for the study were production traits such as total lactation milk yield (LMY), 305-days milk yield (305dMY), 305-days fat yield (305dFY), 305-days solid not fat yield (305dSNFY) and lactation length (LL). Data pruning was done before analysis. Animals with less than 500 kg 305dMY and less than 100 days LL were excluded. Reliability and consistency of pedigree information were checked using pedigree viewer software. After imposing the editing criteria, a total of 809 Sahiwal cows sired by 91 sires were used for final data analysis. Lactations were classified into 6 parities, the maximum parity in the original data set was 12, but due to very few numbers of observation and the higher correlation between parity 6 and later lactations, all parities above 6 were pooled. Likewise, period of birth (or) calving were by 5-year interval.

## Genomic DNA isolation and PCR

Genomic DNAs were isolated from whole blood samples of 150 Sahiwal cows by phenol-chloroform extraction method following standard procedures [16]. The quantity and quality of extracted DNAs were measured by NANODROP 2000 Spectrophotometer (Thermo Scientific, DE, USA). Primers used to amplify the target regions of bovine MASP2 (Accession number: ENSBTAG00000012808) and SIRT1 (Accession number: ENSBTAG00000014023) genes were designed using Primer3Plus (v.0.4.0) online software (http://www.bioinformatics.nl/cgi-bin/primer3plus/primer3plus.cgi) according to *Bos taurus* sequence provided in Ensemble genomic browser (http://www.ensembl.org). The sequences of PCR primers used for amplification, regions, fragment sizes and annealing temperature are shown in Table 1. The PCR reactions were carried out in a total volume of 25 µl on a Thermo-Cycler (Bio-Rad T100) containing 50 ng genomic DNA; 2.0 µl, 0.5 µM of each primer, 13.0 µl of 2X PCR Master Mix and 9.0 µl of nuclease free water. The PCR reaction cycling protocol encompassed initial denaturation at

**Table 1. List of primer sets, its target region, annealing temperatures and amplicon sizes.**

| Gene | Region | Primer sequence | $T_a$ (°C) | Product size (bp) |
|---|---|---|---|---|
| MASP2 | Exon 11 | F: ATCAGGTTTCTGTAAAGCCTCTAT | 61.5 | 689 |
| | | R: ACCACTTCTGGGTCTCATTATCTA | | |
| | Exon 11 | F: GTTTGTGGGAGGAATAGTGTC | 57 | 305 |
| | | R: AGTTAAGTAGTGGAAGAGTGGC | | |
| | Exon 2 & 3 | F: ACAAGTACGCCAACAACCAG | 60.5 | 583 |
| | | R: GCATTGTGATGATGTCAGACC | | |
| SIRT1 | Promotor | F: GTATAGTCCACGGGGTTACAG | 51 | 273 |
| | | R: CCAAACTTGTCTTTCAGAGTC | | |

$T_a$ = Annealing temperature, bp = base pair, °C = degree centigrade

95˚C for 3 min, followed by 34 cycles of 94˚C for 30 s, specific annealing temperature for 30 s, 40 s at 72˚C, and a final extension step at 72˚C for 8 minutes.

## SNP identification and genotyping

Representative samples of PCR products were purified and sequenced by ABI3730XL DNA sequencer (Applied Biosystems, Foster City, CA, USA). Two software programs, CodonCode Aligner (Codon-Code, Dedham, MA) and MEGA11 (Oxford University press), were then used to discover, analyze the sequences and to find the mutation sites and its location. The identified SNPs were further genotyped for all the individuals (n = 150) using polymerase chain reaction-restriction fragment length polymorphism (PCR-RFLP). The amplified PCR products were then subjected to digestion with respective restriction enzyme following manufactures protocol. Subsequently, the digested products were then separated on 2.5 to 2.8% agarose gel and the gel was stained with ethidium bromide. The genotype results of allelic variation were based on the electrophoretic pattern of the restriction enzyme-treated PCR products and then assigned to each individual cows.

## Statistical analysis

**Variance components and genetic parameters.** The statistical significance of fixed effects; season of calving, period of calving and parity, fitted in the model were performed by general linear model using SAS version 9.2 software [17]. Only significant (P<0.05) fixed effects were incorporated into the models, which were subsequently used for genetic analysis. The permanent environmental effect due to the repeated records per animal was taken into account as additional random effects for the analysis of milk production traits (model I), while for the analysis of first lactation milk production traits, permanent environmental effect was not included in the model (model II). Genetic parameters and variance components for the studied traits with repeated records were estimated using single trait repeatability animal model. The general description of the models in matrix forms are given below:

$$Y = X\beta + Zu + Wp + \varepsilon \qquad (I)$$

$$Y = X\beta + Zu + \varepsilon \qquad (II)$$

Where Y is a vector of observed traits; **X**, **Z** and **W** are incidence matrices related to fixed, additive genetic, and permanent environmental effects, respectively. While, **β**, **u**, **p** and **ε** are vector of fixed effects, vector of additive genetic effect, vector of animal permanent environmental effect; and vector of residual effect, respectively. The data were subjected to genetic analysis using BLUPF90 family of programs [18]. The data were renumbered and processed using RENUMF90. The Gibbs sampler was then used to obtain posterior densities of variance components. The marginal posterior distribution for each parameter was obtained by considering one long chain with 1000,000 iterations, where the first 100,000 samples were discarded as burn in and then one out of 50 iterations were stored for further analysis. The convergence of Gibbs chains was monitored through graphical inspection (trace-plots) related to selected parameters. After verifying the graphics, we observed that the burn-in period considered was sufficient to reach convergence in all parameter estimates. Eighteen thousand (18,000) number of effective samples were generated and used to obtain measures of central tendency and the highest posterior density (HPD) region for each parameter. The convergence diagnostic of the chain generated by the Gibbs sampler was then subjected to POSTGIBBSF90 [18], and the HPD region, which provides the interval that includes 95% of samples as a measure of reliability and standard error of parameters were also attained. Computations of variance

components, heritability, repeatability and correlation estimates (genetic, $r_g$; permanent environmental) were carried out using the program GIBBS1F90 and POSTGIBBSF90 [18], respectively.

**Estimation of breeding values and genetic trend.** Individual breeding values of animals for milk production traits were computed from the Best Linear Unbiased Predictions (BLUP) solutions obtained from the repeatability animal model using BLUPF90 [18] software packages. The average EBV of animals were plotted to envisage genetic trend across the birth year. Linear regression coefficients of mean EBV for LMY, 305dMY, 305dFY, 305dSNFY and LL on birth years were computed using REG procedure of SAS to assess the significance of genetic trends in Sahiwal cattle, respectively.

**Association of genotypes with EBVs of milk production traits.** Conventional population genetics statistical analysis: allele frequency, genotype frequency, exact test of Hardy Weinberg equilibrium (HWE) and polymorphism information content (PIC) for each polymorphism were performed using PoPGen2 software [19]. The EBVs of 150 Sahiwal cows were used as a phenotype to test the association of MASP2 and SIRT1 gene SNPs with milk production traits. Thus, the effect of genotypes on the EBVs of milk production traits were analysed using the general linear model (GLM) procedure of SAS 9.2 software [17], with the following model (model III). Moreover, the Tukey-Kramer multiple comparison test was used to analyze significance of differences between groups. Significant level was set to $P \leq 0.05$.

$$\mathbf{Y}_{ij} = \boldsymbol{\mu} + \mathbf{G}_i + \mathbf{e}_{ij} \tag{III}$$

Where, $Y_{ij}$ = Breeding value of $j^{th}$ animal of $i^{th}$ Genotype, $\mu$ is overall mean, $G_i$ is the effect of $i^{th}$ genotypes and $e_{ij}$ is the residual error NID $(0, \sigma^2_e)$.

## Results and discussion

### Identification of SNPs and genotype patterns of different polymorphisms in MASP2 and SIRT1 genes

Study revealed a total of 5 SNPs in bovine MASP2 gene, of which one (g.499C>T) was located in exon 2 region and four (g.576G>A, g.609T>C, g.684G>T and g. 845A>G) were in exon 3 region on chromosome 16 (Table 2). With respect to bovine SIRT1 gene, two SNPs (g.-306T>C and g.-274C>G) were detected in the promoter region of Sahiwal cattle population (Table 2). Further, we individually genotyped three SNPs (g.499C>T, g.609T>C and g.684G>T) in MASP2 gene and two SNPs (g.-274C>G and g.-306T>C) in SIRT1 promoter region while using PCR-RFLP method for 150 Sahiwal cows. Pictorial representation for genotypes is given in Figs 1 and 2.

Restriction digestion using BstUI for MASP2 gene at the SNP locus g.499C>T (rs41255599) produced fragments with lengths of 373 and 210 bp for genotype CC, 583, 373 and 210 bp for genotype CT, and 583 bp for genotype TT (Fig 1C) in Sahiwal cattle population. The second SNP at locus g.609T>C (rs4370651) was obtained with BsrDI enzyme and revealed three genotypes with fragment length of 322 and 261 bp for genotype TT, 583, 322 and 261 bp for genotype TC, and 583 bp for genotype CC (Fig 1C), respectively. Likewise, the third SNP (g.684G>T; rs209711045) in exon 3 of MASP2 gene was obtained with BsrI restriction enzyme and produced three genotypes with three district patterns. These genotypes were GG (583 bp), GT (583, 397, 104 and 82 bp) and TT (397, 104 and 82 bp), respectively (Fig 1C).

For SIRT1 gene the SNP at nucleotide position g.-306T>C locus in promoter region was identified with BsaJI enzyme, resolved in to three polymorphic patterns of TT, TC and CC genotypes in Sahiwal cattle population. The genotype TT represents the occurrence of one band of 273 bp, the genotype TC represents five restriction fragment bands of 273, 197, 39, 29

**Table 2. Names of genes, region and restriction enzymes used for genotyping for the identified SNPs in MASP2 and SIRT1 genes for Sahiwal cattle breed.**

| Gene | Location | SNPs | Mutation | RE | SNP id |
|---|---|---|---|---|---|
| MASP2 | Exon 2 | g.499C>T | C/T | BstUI | rs450770448 |
| | Exon 3 | g.609T>C | T/C | BsrDI | rs4370651 |
| | | g.684G>T | G/T | BsrI | rs209711045 |
| | Promoter | g.-306T>C | T/C | BsaJI | rs718329990 |
| SIRT1 | | g.-274C>G | C/G | SmaI | rs42140046 |

RE = Restriction enzyme, SNPs = Single nucleotide polymorphisms

and 8 bp, and the genotype CC represents four bands of 197, 39, 29 and 8 bp (Fig 2C). In fact, the fragment sizes of 8 and 29 bp is too small, in this respect these fragment sizes were not necessarily observed in the gel. Moreover, SNP locus of g.-274C>G in the promoter region of SIRT1 gene was found in Sahiwal cattle with the enzyme SmaI, that revealed monomorphic patterns in all the studied population. These results differ from the previous findings [12, 13], where three genotypes in this SNP region (g.-274C>G) in Nanyang cattle breed of China and Agerolese cattle breed in Italy, respectively were reported. Interestingly, the polymorphism of these particular mutation site (g.499C>T, g.609T>C, g.684G>T and g.-306T>C) in SIRT1 gene in dairy cattle population is not reported so far. We acknowledge that, our study provides encouragement for a new way to reveal SNPs.

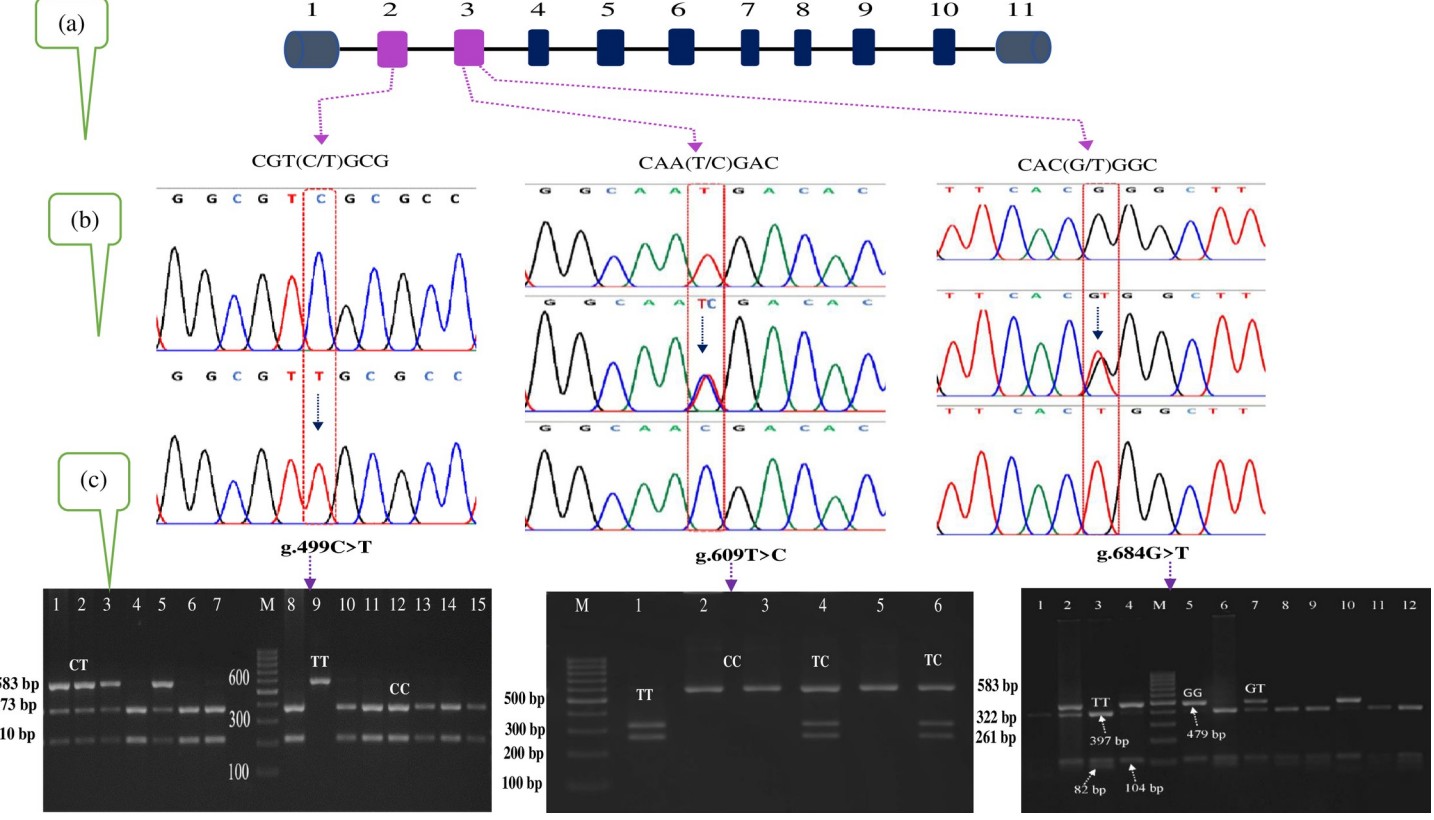

**Fig 1. A schematic diagram of the present study to assess the MASP2 gene polymorphism in the Sahiwal.** (a) Schematic representation of the MASP2 gene with the localization of the three identified SNPs. The blue gray, purple and dark blue bars represent exons, respectively; intervals represent introns. (b) DNA sequencing chromatogram of the polymorphic fragment. (c) PCR-restriction fragment length polymorphism (PCR-RFLP) genotyping of the amplified loci.

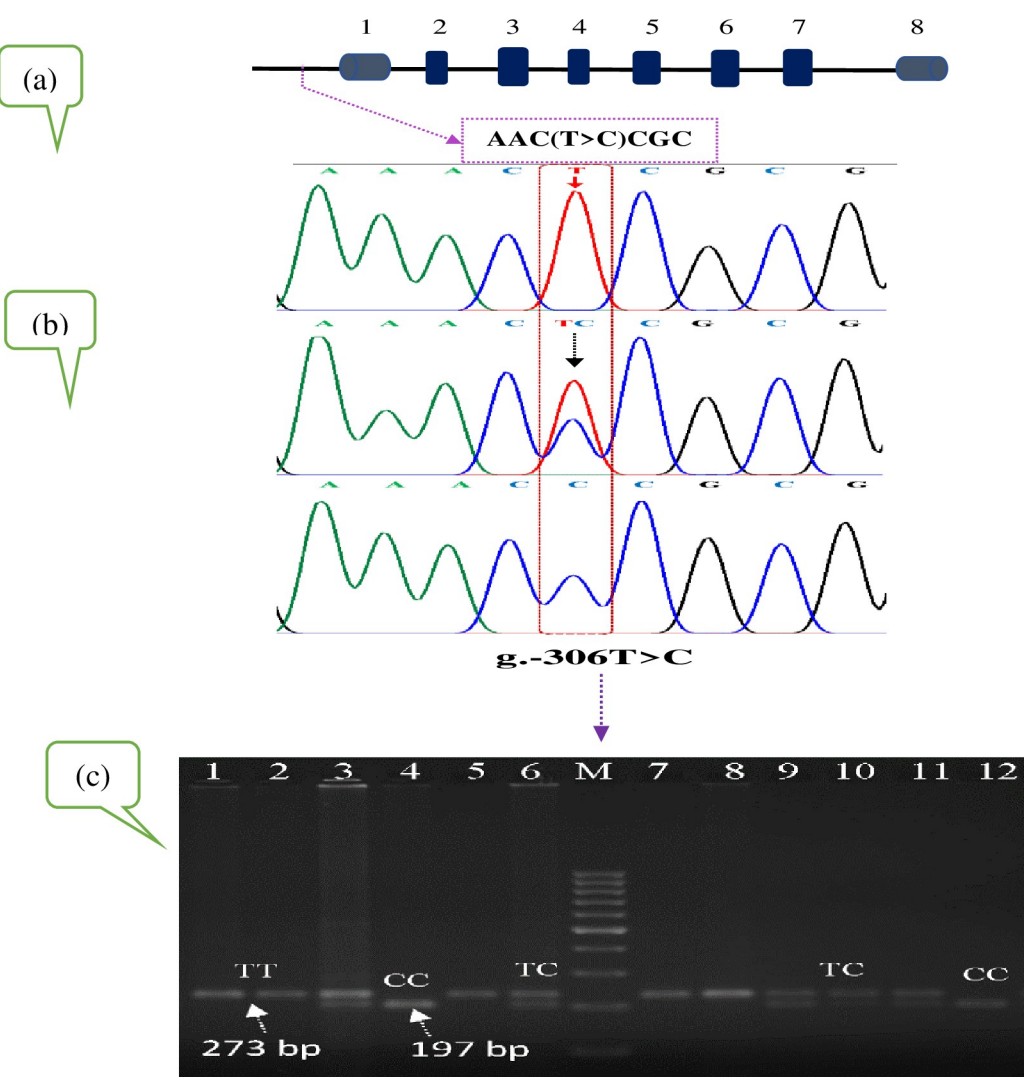

**Fig 2. A schematic diagram of the present study to assess the SIRT1 gene polymorphism in the Sahiwal.** (a) Schematic representation of the SIRT1 gene with the localization of the identified SNP. The line arrow, blue gray and dark blue boxes represent promoter and exons, respectively; intervals represent introns. (b) DNA sequencing chromatogram of the polymorphic fragment. (c) PCR-restriction fragment length polymorphism (PCR-RFLP) genotyping of the amplified locus.

## Allelic and genotypic frequencies of SNPs in MASP2 and SIRT1 genes

The genotype, allelic frequencies, expected heterozygosity, observed heterozygosity, effective population size and its corresponding Chi-squared test to determine whether the population was in Hardy-Weinberg Equilibrium (HWE) or not are presented in Table 3. The level of significance used in the test was 0.05; thus, values above 0.05 indicate that the population was in HWE. Notably, two SNPs (g.499C>T and g.684G>T), the individual frequencies of the genotypes were severely out of Hardy Weinberg equilibrium (HWE). The possible explanation for this could be: (1) Hence Sahiwal is one of the well-established zebu cattle breeds known to have good genetic potential to produce considerably large quantities of milk in India, selection pressure for the increasement of milk production infer the loss of non-favourable alleles. (2) The effect of small population size used for the study cannot be neglected.

**Table 3. The genotype, allelic frequencies and genetic diversity parameters in the MASP2 and SIRT1 genes for Sahiwal cattle.**

| Gene name | SNP | Genotypic frequency | | | Allele frequency | | $(X^2)$ | PIC | He | Ho | ne* |
|---|---|---|---|---|---|---|---|---|---|---|---|
| MASP2 | g.499C>T | CC | CT | TT | C | T | 28.686*** | 0.189 | 0.21 | 0.112 | 1.27 |
| | | 0.824 | 0.112 | 0.064 | 0.88 | 0.12 | | | | | |
| | g.609T>C | TT | TC | CC | T | C | 2.452 | 0.260 | 0.31 | 0.264 | 1.44 |
| | | 0.06 | 0.26 | 0.68 | 0.19 | 0.81 | | | | | |
| | g.684G>T | GG | GT | TT | G | T | 13.84*** | 0.298 | 0.36 | 0.248 | 1.58 |
| | | 0.12 | 0.25 | 0.63 | 0.24 | 0.76 | | | | | |
| SIRT1 | g.-306T>C | TT | TC | CC | T | C | 0.391 | 0.368 | 0.49 | 0.521 | 1.95 |
| | | 0.32 | 0.52 | 0.16 | 0.58 | 0.42 | | | | | |

*** = Significant at P < 0.0001, SNP = Single nucleotide polymorphism, $X^2$ = Chi-square, PIC = Polymorphic information content, He = Expected heterozygosity,

Ho = Observed heterozygosity,

ne* = Effective number of alleles

At SNP g.499C>T locus in MASP2 gene, the frequency of CC genotypes (0.82) was higher than CT (0.11) and TT (0.06) genotype in Sahiwal cows, respectively. Though, C allele was the most abundant in Sahiwal cattle population under the study (Table 3). Likewise, SNP at nucleotide position g.609T>C, the corresponding genotypic frequencies were 0.06 (TT), 0.26 (TC) and 0.68 (CC), respectively. At this locus the T allele frequency was minimal. Similarly, at locus g.684G>T, the frequency of TT genotype (0.63) was higher than GT (0.25) and GG (0.12) genotypes. The T allele frequency was significantly higher compared to its counterparts G allele in the studied population (Table 3). Further analysis for SNP g.-306T>C of SIRT1 gene indicated that TC (0.52) genotype was higher than TT (0.32) and CC (0.16) genotypes, respectively.

The genetic indices of gene homozygosity (Ho), gene heterozygosity (He), polymorphic information content (PIC) and $X^2$ values are effective to assess the genetic diversity from different loci of candidate genes. PIC indicates the possibility of a marker allele derived from the same allele of its father or mother [12]. According to the results of the present study, the PIC and He value of the identified SNPs exhibited that, the genetic diversity of Sahiwal cattle were low (PIC value < 0.25) to medium polymorphism level (0.25<PIC<0.50). The low genetic diversity in Sahiwal breed in this study is a result of selective breeding, where this herd is being selected for several generations for the milk yield. However, frequent external genetics is introgressed from the field on a regular time interval to maintain the desired heterozygosity. Moreover, the studied population possess medium to high effective allele number (ranging from 1.27 to 1.95). This indicates that these SNPs have a potential for selection. As a matter of fact, many of the SNPs identified in this study have not been reported earlier in dairy cattle. Given the evidence of very little information on these SNPs, this study failed to provide adequate proof from the literature.

## Associations of genotypes of MASP2 and SIRT1 genes with EBV of milk production traits

Association analysis of SNPs (variants) of candidate genes is a needful step for the knowledge of the genetic basis of economically important traits, and compared to other genomic approaches is potentially more easily and efficiently implemented in the breeding program [20]. Table 4 shows the association analysis of MASP2 and SIRT1 genotypes upon estimated breeding values (EBVs) for milk production traits. The association analysis of the present study revealed that, in the MASP2 gene, two SNPs, g.499C>T and g.609T>C, remained significantly associated with

**Table 4. Association of different genotypes of SNPs in MASP2 and SIRT1 genes with EBV of production traits for Sahiwal cattle.**

| Gene | Loci | Genotype | LMY | 305dMY | 305dFY | 305dSNFY | LL |
|---|---|---|---|---|---|---|---|
| MASP2 | g.499C>T | CC (102) | 0.22[b] | 5.43[b] | -0.52[b] | -0.27[b] | 0.97 |
| | | CT (13) | 73.98[ab] | 69.43[ab] | 0.21[ab] | 0.93[ab] | 8.67 |
| | | TT (8) | 195.85[a] | 180.20[a] | 1.30[a] | 2.88[a] | 15.47 |
| | | P | 0.011 | 0.013 | 0.025 | 0.031 | 0.060 |
| | g.609T>C | TT (7) | -188.60[b] | -164.19[b] | -2.49[b] | -4.03[b] | -7.44 |
| | | TC (32) | 35.38[a] | 36.97[a] | -0.30[a] | 0.33[a] | 3.63 |
| | | CC (84) | 33.00[a] | 34.45[a] | -0.15[a] | 0.29[a] | 3.26 |
| | | P | 0.011 | 0.012 | 0.010 | 0.005 | 0.356 |
| | g.684G>T | GG (15) | -23.93 | -15.65 | -0.76 | -0.70 | -1.03 |
| | | GT (30) | 64.00 | 59.56 | -0.29 | 0.21 | 4.90 |
| | | TT (78) | 13.02 | 17.41 | -0.25 | 0.15 | 2.63 |
| | | P | 0.301 | 0.346 | 0.652 | 0.672 | 0.625 |
| SIRT1 | g.-306T>C | TT (29) | -47.44[b] | -33.60[b] | -0.56[b] | -0.50[b] | -3.13[b] |
| | | TC (48) | 120.40[a] | 109.66[a] | 0.64[a] | 1.61[a] | 8.63[a] |
| | | CC (15) | -4.58[ab] | -1.38[ab] | -0.39[ab] | -0.17[ab] | 2.84[ab] |
| | | P | 0.0006 | 0.0013 | 0.0180 | 0.0219 | 0.0423 |

LMY = Total lactation milk yield, 305dMY = 305-days milk yield, 305dFY = 305-days fat yield, 305dSNFY = 305-days solid not fat yield, LL = Lactation length, means with different superscript differs significantly, P = P value

EBV of LMY, 305dMY, 305dFY and 305dSNFY ($P < 0.0058$–0.0256) in Sahiwal cows, respectively. Interestingly, the TT genotype of SNP g.499C>T locus possessed the highest EBV for all milk yield traits, indicating that the mutant allele T was associated with superior milk performances in the studied population. On the other hand, CC genotyped cows associated with the lowest EBV for the traits studied. While, the EBV of CT genotyped cows found to be intermediate as compared to TT genotypes. For the SNP change at position g.609T>C, the TC and CC genotyped cows had significantly higher estimated breeding value of the studied traits as compared to TT genotype, indicating that allele C was associated with superior genetic merit for milk yield traits (Table 4). However, these two SNPs did not show significant effect on the EBV of LL (P values > 0.05) of Sahiwal cows. Curiously, this SNP locus could become a crucial molecular and genetic marker as it showed a significant impact on all the EBVs of milk yield traits under the study. These results offer overwhelming evidence to the research in to genetic markers, in particular markers associated with the improvement of milk yield traits in Zebu cattle breeds. The EBV of GT genotyped cows at the SNP g.684G>T locus appeared to be high in magnitude, but statistically not significant ($P < 0.05$). No studies about the association of these specific SNPs of bovine MASP2 gene on various traits in livestock (especially in cattle) were available, so that evidences have not been drawn from the literature.

In the present study, the association analysis revealed that SNP g.-306T>C in SIRT1 promoter region was strongly associated ($P<0.0006$–0.0423) with EBVs of LMY, 305dMY, 305dFY, 305-dSNF and LL in Sahiwal cattle population. To put it in another way, EBVs of TC genotype cows was significantly higher than those of CC and TT genotype cows, respectively (Table 4). In case of SIRT1 gene, TT genotype cows demonstrated the lowest lactation performance in Sahiwal cattle. In this regard, our results confirmed that the priority of genomic and inbreed selection should be held on heterozygous states as compared to homozygous ones. The heterozygote TC is phenotypically different from the homozygote TT and CC because it has a higher value for a production trait or has a phenotype thought to be important for the

studied breed, then the heterozygote TC can have a selective advantage due to artificial selection. To the best of our knowledge, no other studies have found this specific SNP g.-306T>C and its impact on economically important traits in dairy cattle breeds. In this study, we provide the first evidence for the significant association of the g.-306T>C variant with milk yield traits in dairy cattle.

Mannose binding lectin associated serine protease 2 (MASP2) is the central protease in the complement system. The complement lectin pathway is an important component of the innate immunity, in which MASP1 is one of the central proteases. Coupled with this, earlier study [21] reported that the lectin pathway might be activated in subjects with chlorine-esterase-inhibitor deficiency, which is linked with low MASP-2 and complement 4 levels. Previous study [14] identified the SNP (G553A) in the third exon (CUB1 domain) of bovine MASP2 gene and investigated the significant association of this SNP and somatic cell score in Chinese Holstein cattle. Moreover, three other SNPs (g.14047A>C, g.14248T>C and g.14391C>T) in the eleventh and 3'UTR region of MASP2 gene are also reported [15]. Accordingly, they reported that the SNPs g.14047A>C and g.14248T>C were significantly associated with protein percentage and 305-day milk yield, respectively. Specifically, they mentioned that, cows with genotype TT had higher 305-day milk yield than those with genotype of TC of g.14248T > C locus. However, here in our study, we did not find these SNPs, but three other SNPs (g.499C>T and g.609T>C and g.684G>T) in exon 2 and 3 region were explored (Fig 2). In light of all these, the results from this study suggests that, the SNPs g.499C>T (rs450770448) and g.609T>C (rs4370651) in MASP2 gene can be a useful candidate gene for milk production traits and would be applied in marker assisted selection in dairy cattle. However, the results need to be interpreted with caution, hence the favourable genotypes are based up on relatively small number of observations, much larger sample sizes are needed to obtain a reasonable number of TT genotypes for critical comparison.

Mammalian SIRT1gene has evolved to modify the activity of a growing number of transcription factors, including P53, NF-κB, and PGC-1α, suggesting that SIRT1 functions in a wide range of cellular responses to stress, inflammation, and nutrients. Until now, in bovine species, only few studies in silent information regulator 1 gene polymorphisms in dairy animals have been reported in the literature. Earlier study [12] explored the presence of 5 SNPs, of which g.-382G>A and g.-274C>G located in the promoter region, while g.17324T>C, g.17379A>G and g.17491G > A located in noncoding regions of the SIRT1 gene. Subsequently, some studies [22, 23] also examined many noncoding mutations in the 3'UTR region of the SIRT1 gene (g.25764G>A, g.25846A>G, g.25868T>C, g.25751A>C). Some of the above-mentioned SNPs significantly influenced various growth and carcass traits in Nanyang, Qinchuan and Luxi cattle breeds [12, 22, 24]. Even though the previous studies in SIRT1 gene relied on various growth and carcass traits, other work [13] reported the significant effect of g.-274C>G locus for milk production and reproductive performance traits in Agerolese cattle breed. In this study, we detected two SNPs (g.-306T>C and g.-274C>G) in the promoter region of the gene, nonetheless, the SNP g.-274C>G found to be monomorphic in the studied Sahiwal cattle population. In this study, we provide the first evidence for significant association of the g.-306T>C variant with milk yield traits in dairy cattle.

## Estimation of genetic parameters for milk production traits

**Sizable additive genetic variance suggest scope of selecting cows for lactation traits.** Estimates of genetic parameters are necessary to determine the selection criterion and future breeding strategies. Estimated posterior means of variance components, heritability and repeatability of the studied traits are presented in Table 5. The posterior mean of heritability

**Table 5. Estimates of posterior means of variance components and genetic parameter estimates for milk production traits in Sahiwal cattle.**

| Traits | $\sigma^2_a$ | $\sigma^2_{pe}$ | $\sigma^2_e$ | $\sigma^2_p$ | $h^2\pm$PSD | $r\pm$PSD |
|---|---|---|---|---|---|---|
| Whole lactation (1st parity-6th parity) | | | | | | |
| LMY | 10950 | 35043 | 46527 | 92520 | 0.12±0.04 | 0.50±0.02 |
| 305dMY | 9010 | 28279 | 40471 | 77759 | 0.115±0.04 | 0.48±0.02 |
| 305dFY | 22.9 | 85.86 | 303.7 | 412.56 | 0.06±0.03 | 0.26 ± 0.03 |
| 305dSNF | 78.8 | 346.64 | 1001.5 | 1426 | 0.06 ± 0.03 | 0.30±0.03 |
| LL | 305.5 | 372.4 | 3631.6 | 4309.5 | 0.07± 0.02 | 0.16±0.02 |
| First lactation (1st parity) | | | | | | |
| LMY | 18054 | - | 72795 | 90849 | 0.20±0.08 | - |
| 305dMY | 8438.4 | - | 42683 | 51122 | 0.17±0.08 | - |
| 305dFY | 150.63 | - | 990.81 | 1141.4 | 0.13±0.09 | - |
| 305dSNFY | 511.20 | - | 3394.9 | 3906.1 | 0.13±0.09 | - |
| LL | 1861.7 | - | 5977.2 | 7838.8 | 0.24±0.07 | - |

LMY = Total lactation milk yield, 305dMY = 305-days milk yield, 305dFY = 305-days fat yield, 305dSNFY = 305-days solid not fat yield, LL = Lactation length, $\sigma^2_a$ = Additive genetic variance, $\sigma^2_{pe}$ = Permanent environment variance, $\sigma^2_e$ = Residual variance, $\sigma^2_p$ = Phenotypic variance, $h^2$ = Heritability, PSD = Posterior standard deviation, r = Repeatability

estimates for milk production traits ranged between 0.13 and 0.24 for first lactation traits and 0.06 and 0.12 for whole lactation traits (repeatability model), respectively (Table 5). The estimates were better for the first parity and relatively less in magnitude for all parities combined together using repeatability model. Relatively better $h^2$ estimates for first lactation milk production traits were observed as the data for first parity is free of selection and culling bias. The decline of $h^2$ estimates in all of the analyzed traits from whole lactations may also be attributed to a large increase in permanent environmental variance. The moderate heritability estimates for first lactation milk production traits (LMY: 0.20±0.08, 305dMY: 0.17±0.08, 305dFY: 0.13

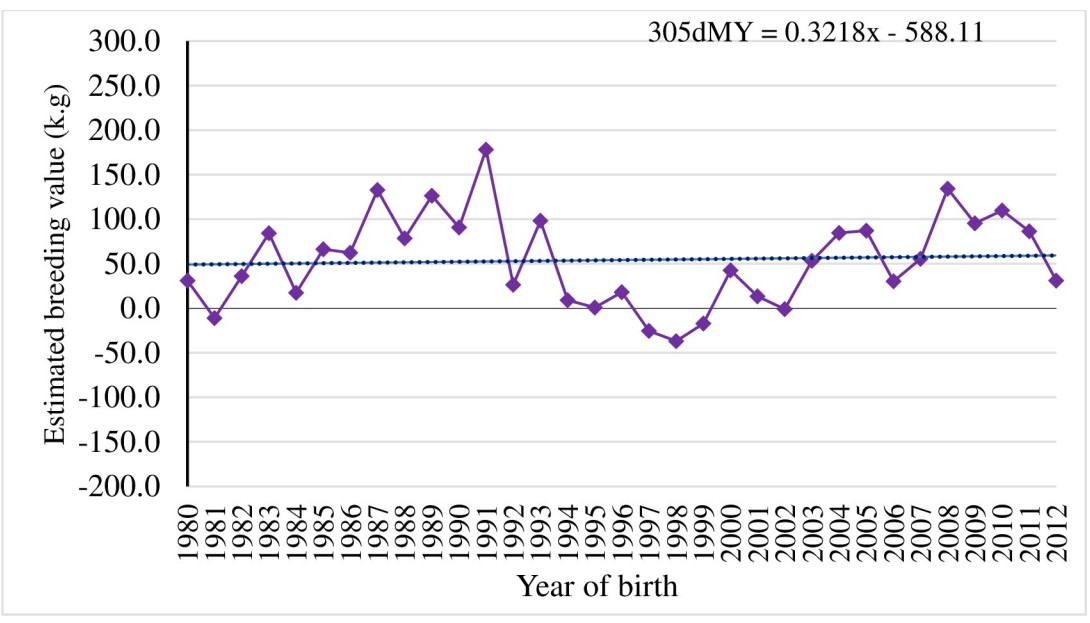

**Fig 3. Genetic trend for 305-days milk yield (305dMY) by year of birth of Sahiwal cows.**

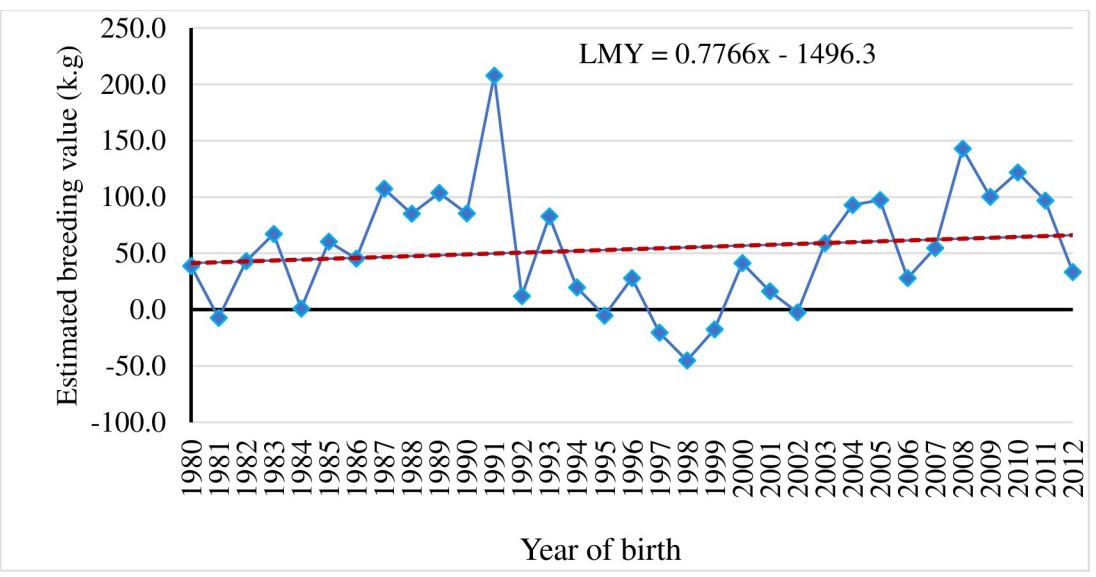

**Fig 4. Genetic trend for total lactation milk yield (LMY) by year of birth of Sahiwal cows.**

±0.09, 305dSNFY: 0.13±0.09 and LL: 0.24±0.07) indicates the presence of sufficient genetic variation and scope for further improvement through genetic selection.

Low heritability for LMY (0.16) and LL (0.07) in Sahiwal was also reported in semi-arid Kenya [25]. Similar low heritability estimates for lactation milk yield (0.14±0.03), 305-days milk yield (0.13±0.04) and lactation length (0.074±0.028) for Frieswal cattle were also reported [26]. Heritability of 0.28 for 305 days or less milk yield, 0.21 for first lactation total milk yield and 0.26 for first lactation length were observed in Sahiwal cattle [27]. In contrast to our results, low heritability estimates of 0.059 in the first lactation, 0.083 in the second lactation and 0.052 in the third lactation for fat yield were also noticed in Holstein cows [28]. While, higher heritability estimates were reported for milk yield and fat yield of 0.29 and 0.34 for Chilean cattle breed and 0.30 and 0.32 for Jersey dairy cattle in Zimbabwe, respectively were

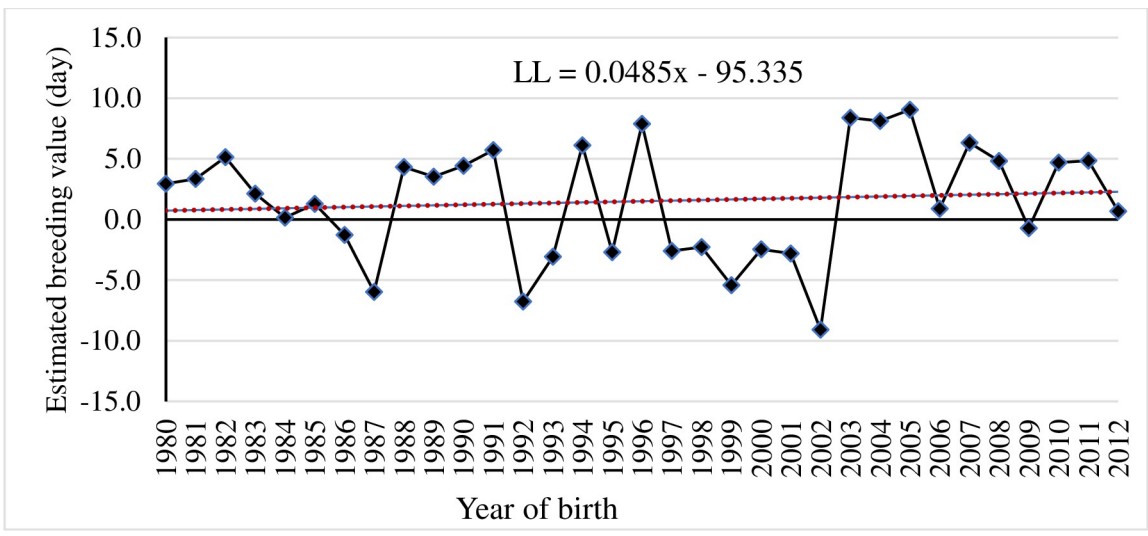

**Fig 5. Genetic trend for lactation length (LL) by year of birth of Sahiwal cows.**

**Table 6. The posterior means of genetic correlations with posterior standard deviation (PSD) of whole lactation (above diagonal) and first lactation (below the diagonal) for production traits using a two-trait model for Sahiwal cattle.**

| Traits | LMY | 305dMY | 305dFY | 305dSNFY | LL |
|--------|-----|--------|--------|----------|-----|
| LMY | | 0.99±0.003 | 0.99±0.006 | 0.99±0.006 | 0.92±0.04 |
| 305dMY | 0.99±0.001 | | 0.99±0.005 | 0.99±0.003 | 0.91±0.05 |
| 305dFY | 0.97±0.006 | 0.98±0.001 | | 0.98±0.03 | 0.55±0.40 |
| 305dSNFY | 0.97±0.004 | 0.99±0.004 | 0.98±0.002 | | 0.77±0.20 |
| LL | 0.92±0.006 | 0.94±0.006 | 0.83±0.01 | 0.86±0.01 | |

LMY = Total lactation milk yield, 305dMY = 305-days milk yield, 305dFY = 305-days fat yield, 305dSNFY = 305-days solid not fat yield, LL = Lactation length

reported [29, 30]. Looking at the results for first lactation traits in our study, we infer that the genetic selection through first lactation traits will have the opportunity to improve these traits in the herd for further improvement.

**Repeatability estimates for lactation traits suggest consistent performance of cows over the parities.** Knowing the repeatability estimates of economic traits are beneficial in making management and breeding decisions, since repeatability estimates are used to predict future performance from past records [31]. In this study, the posterior mean of repeatability estimates for LMY, 305dMY, 305dFY and 305dSNFY were 0.50±0.02, 0.48±0.02, 0.26±0.03 and 0.30 ±0.03, respectively (Table 5). These moderate to high repeatability estimates obtained in the present study signified that in the Sahiwal cows, future performance can be predicted accurately from early first lactation records. Thus, culling Sahiwal cows on the basis of early records for milk yield traits would be of high importance. However, low repeatability estimates for LL (0.16±0.02) in this study is indicative of dominance of temporary environment for LL determination. The repeatability estimates of LMY and 305dMY of the present study coincides with the estimates obtained in the previous studies [25, 29, 32] for Sahiwal, Frieswal and Chilean dairy overo Colorado breed, respectively. Repeatability of 0·42 for milk yield traits and 0·31 for lactation length were reported for Sahiwal cattle in Pakistan [33]. Repeatability of 0.39 for milk yield and 0.38 for fat yield of Jersey cattle in Zibabwe was also reported in literature [30]. Another author [34] reported lower repeatability estimate of 0.33 and 0.10 for LMY and LL for Jersey cattle, respectively. Furthermore, repeatability estimates of 0.475, 0.847 and 0.898 for milk yield and 0.198, 0.281 and 0.279 for fat yield in the first three lactations of Holstein cows was also reported [28]. Differences in estimates are mostly due to breed differences. In the present study, the estimates of repeatability were extremely higher than the corresponding heritability, indicating presence of much higher impact of permanent environment in the population. As justified [1], the evidence we found points to the reason that the higher repeatability estimate is due to non-transmittable effects, permanent environment, and non-additive genetic effects common to all lactations.

**Realized genetic gain for milk production traits in Sahiwal cattle was low.** The genetic trend of all the studied milk production traits were estimated using regression of means of EBVs over year of birth. The yearly EBVs for 305dMY, LMY and LL are depicted in Figs 3–5. Genetic trend revealed meagre magnitude of genetic progress in milk production traits over the years in Sahiwal cattle (P>0.05). There was selection program taking place in the herd, however as the generations are overlapping, it is difficult to see this progress (genetic progress over generation not shown here). Moreover, there are a lot of purchasing of animals and involuntary culling of good animals too. In any genetic improvement program, there is a need of tracking the results to evaluate the genetic progress achieved, to make adjustments aiming to optimize genetic gain, and to increase farm profitability in the future [35]. In general, EBVs for

305dMY, LMY and LL were increased by 0.32 kg, 0.77 kg and 0.04 day/year in Sahiwal cattle population, although the trends were non-significant. Given the fact that the estimated heritability for milk production traits obtained in the present study were low, the genetic changes for these traits remained negligible. The meagre change in response could be due to low intensity of selection and huge environmental influence on the studied traits such as significant fluctuation in performance over the years. Culling of high genetic merit animals attributed to involuntary reasons such as mastitis or reproductive infection also affects the genetic progress negatively as they cannot contribute sufficient progeny to the next generation. Similar to this study, [33, 36] reported adverse impact of negative environmental factors on trends for milk production and fertility traits in Sahiwal herds in Pakistan. Conversely, favourable genetic progress has been reported for milk yield in some Sahiwal herds in India [37], in Jersey [38] and Holstein-Friesian cattle breeds in Kenya [39]. Higher genetic trends for milk yield (80 kg per year) and fat yield (5 kg per year) of Brown Swiss cattle in the tropics of Mexico [40] is also reported. Similarly, positive genetic progress for milk yield of Jersey cows in South Africa [41] is also reported.

## Favourable genetic correlation of first with later parity lactation traits suggestive of selecting Sahiwal cows using first parity records

The estimates of genetic correlations among milk production traits in Sahiwal cattle are presented in Table 6. The genetic correlations between milk production traits obtained in this study were generally high and ranged from 0.83 to 0.99 in first lactation and 0.55 to 0.99 in whole lactation milk production traits, respectively. Phenotypic correlations (not shown) were almost similar with the corresponding genetic correlations (values ranged from 0.63 to 0.99). The existing strong positive and desired genetic correlations among production traits indicates the efforts for improvement in one trait will lead to an improvement in the other trait. This is due to the pleiotropic effect, where; different traits are influenced by the same set of genes. Similar to our finding, several studies reported positive and strong genetic correlations among milk yield traits, ranging from 0.49 to 0.92 [26, 29, 30, 42, 43]. The genetic correlation between LMY and LL reported in this study is consistent with those reported in Kenya [25] and in Pakistan [33] for Sahiwal cattle, thus cows that produce more LMY will be milked for prolonged periods. Our result illustrates that milk production traits are influenced by the same set of genes in the subsequent lactations and therefore collection of first lactation records should be sufficient enough for the routine genetic evaluation of these milk production traits in Sahiwal cattle population. These results are in agreement with earlier studies [44, 45] which concluded that dairy performance of cows in all lactations are determined by more or less the same set of genes, and first parity milk yield is therefore considered as an efficient selection criterion for lifetime production. The high and desired genetic correlation between first lactation traits and the moderate heritability in the first lactation for both milk production traits suggest that collection of first lactation performance records, especially 305dMY should be sufficient for routine genetic evaluation, without much expenditure for milk data collection and analysis in later lactations for the Sahiwal cows.

## Conclusion

Indian Sahiwal cattle are the fastest growing breed with good milk production potential and largely exploited for the production of many dairy products in India. For this reason, the improvement of selection methods is of striking economic relevance while a deeper understanding of the genetic mechanisms affecting milk production trait is a general scientific interest. Our study shows evidence of a strong significant associations between SNP within the

MASP2 (g.499C>T and g.609T>C) and SIRT1 (g.-306T>C) gene and milk production traits in Sahiwal cows, indicating the potential role of MASP2 and SIRT1 variants in these traits. After validation on large sample size population, the identified alleles could be included in the selection program. In this study, milk production traits had better heritability estimate for the first lactation traits compared to the estimates from the whole lactations. Therefore, a high response to selection in first lactation traits would be expected in this population. The repeatability estimates for milk yield traits were medium to high, implying that selection of cows based on early performance is possible. Strong positive and desired genetic correlation of 305dMY with other lactation traits in first parity indicates that the Sahiwal cows can be further selected using 305dMY trait only in the first parity, which will bring overall improvement in the other lactation traits and will reduce cost on resource management. Taken together, a 2-stage selection approach, select Sahiwal animals early for the SNP and then on the basis of 30dMY will help to improve genetic potential and also save resources.

## Supporting information

**S1 File. Parameter file for genetic parameter estimation of milk production traits in Sahiwal cattle using single trait Gibbs sampling repeatability animal model.**
(PDF)

**S2 File. Parameter file for genetic parameter estimation of first lactation traits in Sahiwal cattle using single trait Gibbs sampling animal model.**
(PDF)

**S3 File. Estimated breeding value data of milk production traits for Sahiwal cattle.**
(XLSX)

**S1 Raw image. Agarose gel electrophoresis (1.5%) of PCR products of Exon 2 & 3 in MASP2 gene.** The original gel image data was captured by a UV trans illuminator gel documentation system and saved as tiff file. Lane 1–33: PCR product (583 bp). Lane M1, M2: 100 bp DNA marker.
(PDF)

**S2 Raw image. Agarose gel electrophoresis (1.7%) of PCR products of promoter region of SIRT1 gene.** The original gel image data was captured by gel documentation system (Gel. LUMINAX) and saved as tiff file. The lanes marked with "X" are non-specific amplifications and these experimental samples are not included in further genotyping purpose. Lane 2–17: PCR product (273 bp). Lane M: 100 bp DNA marker.
(PDF)

**S3 Raw image. Raw gel images of PCR-RFLP genotyping of the SNPs g.499C>T, g.609T>C and g.684G>T within MASP2 gene in Fig 1C.** (A) The PCR products of MASP2 gene at the SNP g.499C>T locus in Fig 1C was digested with BstUI enzyme and the digested products were separated by 2.8% agarose gel electrophoresis stained with ethidium bromide. The original gel image data was captured by a gel documentation system (Gel.LUMINAX) and saved as tiff file. The fragment sizes of the identified genotypes are shown in the left (583, 373 and 210 bp). Three genotypes were detected in this experimental population, namely CC (lanes 4, 6, 7, 8, 10–15), CT (lanes 1, 2, 3, 5) and TT (lane 9). Lane M is the 100 bp marker. The lanes not included in the final Fig 1C marked with "X" above the lane label on the original gel image. The final figure panel generated from the original image is shown in Fig 1C (left). (B) The PCR products of MASP2 gene at the SNP g.609T>C locus in Fig 1C was digested with BsrDI enzyme and the digested products were separated by 2.8% agarose gel electrophoresis

stained with ethidium bromide. The original gel image data was captured by a gel documentation system (Gel.LUMINAX) and saved as tiff file. The fragment sizes of the identified genotypes are shown in the right (583, 322 and 261 bp). Three genotypes were detected in this experimental population, namely TT (lane 1), TC (lanes 4, 6) and CC (lanes 2, 3, 5). Lane M is the 100 bp marker. The lanes not included in Fig 1C marked with "X" above the lane label on the original gel image. The final figure panel generated from the original image is shown in Fig 1C (middle). (C) The PCR products of MASP2 gene at the SNP g.684G>T locus in Fig 1C was digested with BsrI enzyme and the digested products were separated by 2.8% agarose gel electrophoresis stained with ethidium bromide. The original gel image data was captured by a gel documentation system (Gel.LUMINAX) and saved as tiff file. Three genotypes, GG (lane 5), GT (lanes 2, 4, 7, 10) and TT (lanes 1, 3, 6, 8, 9, 11, 12) were detected in this experimental population. Lane M is the 100 bp marker. The lanes not included in Fig 1C marked with "X" above the lane label on the original gel image. The final figure panel generated from the original image is shown in Fig 1C (right).
(PDF)

**S4 Raw image. Raw gel image of PCR-RFLP genotyping of the SNP g.-306T>C within SIRT1 gene in Fig 2C.** The PCR products of the SIRT1 gene at the SNP g.-306T>C locus in Fig 2C was digested with BsaJI enzyme and the digested products were separated by 3% agarose gel electrophoresis stained with ethidium bromide. The original gel image data was captured by a gel documentation system (Gel.LUMINAX) and saved as tiff file. Three genotypes were detected in this experimental population, namely TT (lanes 1, 2, 5, 7, 8), TC (3, 6, 9–11) and CC (lanes 4, 12). For TC and CC genotypes, the fragment sizes of 39, 29 and 8 bp are invisible in the figure. Lane M is the 100 bp marker. The lanes not included in the final Fig 2C marked with "X" above the lane label on the original gel image. The final figure panel generated from the original image is shown in Fig 2C.
(PDF)

## Acknowledgments

Authors would like to acknowledge the Director, ICAR-National Dairy Research Institute, Head, Dairy Cattle Breeding Division for providing necessary facilities in the lab for carrying out the above work.

## Author Contributions

**Conceptualization:** Destaw Worku, Gopal Gowane, Archana Verma.

**Data curation:** Destaw Worku, Gopal Gowane.

**Formal analysis:** Destaw Worku.

**Investigation:** Destaw Worku, Gopal Gowane, Archana Verma.

**Methodology:** Destaw Worku, Gopal Gowane.

**Resources:** Archana Verma.

**Software:** Destaw Worku.

**Supervision:** Gopal Gowane, Archana Verma.

**Validation:** Destaw Worku, Gopal Gowane, Archana Verma.

**Visualization:** Destaw Worku, Gopal Gowane, Rani Alex, Pooja Joshi, Archana Verma.

**Writing – original draft:** Destaw Worku.

**Writing – review & editing:** Destaw Worku, Gopal Gowane, Rani Alex, Pooja Joshi, Archana Verma.

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
