## [Decision Letter · Decision Letter 0]

26 Dec 2021

PONE-D-21-30537Inputs for optimizing selection platform for milk production traits of dairy Sahiwal cattlePLOS ONE

Dear Dr. Destaw Worku,

Thank you for submitting your manuscript to PLOS ONE. After careful consideration, we feel that it has merit but does not fully meet PLOS ONE’s publication criteria as it currently stands. Therefore, we invite you to submit a revised version of the manuscript that addresses the points raised during the review process.Indicate which changes you require for acceptance versus which changes you recommendAddress any conflicts between the reviews so that it's clear which advice the authors should followProvide specific feedback from your evaluation of the manuscriptPlease submit your revised manuscript by 15.01.2022. If you need more time than this to complete your revisions, please reply to this message or contact the journal office at plosone@plos.org. Please include the following items when submitting your revised manuscript:A rebuttal letter that responds to each point raised by the academic editor and reviewer(s). You should upload this letter as a separate file labeled 'Response to Reviewers'.A marked-up copy of your manuscript that highlights changes made to the original version. You should upload this as a separate file labeled 'Revised Manuscript with Track Changes'.An unmarked version of your revised paper without tracked changes. You should upload this as a separate file labeled 'Manuscript'.

We look forward to receiving your revised manuscript.

Kind regards,

Md Ashrafuzzaman, Ph.D.

Academic Editor

PLOS ONE

Journal Requirements:

In your cover letter, please note whether your blot/gel image data are in Supporting Information or posted at a public data repository, provide the repository URL if relevant, and provide specific details as to which raw blot/gel images, if any, are not available. Email us at plosone@plos.org if you have any questions

Reviewers' comments:

Reviewer's Responses to Questions

**Comments to the Author**

1. Is the manuscript technically sound, and do the data support the conclusions?

Reviewer #1: Partly

Reviewer #2: Yes

2. Has the statistical analysis been performed appropriately and rigorously? 

Reviewer #1: Yes

Reviewer #2: Yes

3. Have the authors made all data underlying the findings in their manuscript fully available?

Reviewer #1: Yes

Reviewer #2: Yes

4. Is the manuscript presented in an intelligible fashion and written in standard English?

Reviewer #1: Yes

Reviewer #2: Yes

5. Review Comments to the Author

Reviewer #1: Abstract: Ok

This is a good piece of work. A comprehensive study on exploring the genetic architecture of milk production traits in dairy cattle is of great importance in livestock industry. However, if the following issues are addressed carefully, this work could have a great impact in improving productivity and save costs in dairy industry.

I suggest the author should undergo thorough revision (major) of the manuscript.

Introduction:

Regarding justification of the experiment, since the author mentioned (line number 101-103) very few studies have reported the genetic association of the SNP variation in MASP2 and SIRT1 gene with milk production traits. However, it would have been a novel work, if there was no previous report of association of these two genes with milk production traits.

Materials and method:

In line number 192, the author mentioned nine thousand and within the parenthesis mention 18000, which one is correct?

Results and discussion:

Line number 226: “in exon 3 region of” please replace ‘of’ with on.

Line number 233: “BstUI for MASP2 SNP at the locus” please change the line as “BstUI for MASP2 gene at the SNP locus”…………………

Line number 275: genetic diversity parameters and its corresponding Chi-squared test to determine….

Please mention the name of genetic diversity parameters such as expected heterozygosity, Observed heterozygosity, effective population size…

Line number 288-289: “At this locus the T allele frequency was minimal” which contradicts with that given in Table:3. In table 3, T allele is shown to have allele frequency of 0.81 for T allele and 0.19 for the allele C. The allele C’s frequency would be 0.81 and T allle’s frequency would be 0.19.

In line 299: The author described the genetic diversity in Shahiwal cattle as low PIC and low He indicating low genetic diversity. How will you explain the cause of low genetic diversity/ loss of genetic diversity in Sahiwal cattle? If the loss in diversity continues in this breed, this breed will face extinction. How would you solve this? Or what is your suggestion to improve the diversity? Please write in brief.

Line number 342: “Association analysis of candidate gene is a needful…..” Please rewrite or change like:

Association analysis of SNPs (variants) of candidate genes is a needful …………………….

Line 359-360: I find no reason to mention about the statistically non-significant SNPs in the text since it has been shown in the table. Please remove part of line 359-360…

Line 369: “In particular, TT genotype cows”………Please change to “In case of SIRT1 gene, TT genotype cows……

Line 370-371. It is suggested by the author to select heterozygotes (TC) for SNP, g-360T>C allele? How would you explain this higher performance of TC than CC and TT? Please write the explanation.

Line number 460-462: We know, usually, the repeatability value of a trait is higher or equal to the heritability values due to the differences in the calculation formula. In repeatability calculations we account for the permanent environmental effect. Therefore, higher repeatability value obtained in the present study might be attributed to much higher impact of permanent environment in the population. Please rewrite the sentences…………..

Line number 482, Please rewrite the sentence as : in general, EBVs for 305dmY, LMY and LL were increased………………

Line 485-486: The meagre change in response is due to low-selection intensity………………is in contrast to the line 280-282, where the author explained the reason for loss of genetic diversity due to increased selection pressure. Please rewrite the sentences and explain carefully.

Line 482: would you please explain the reason for very low genetic progress in LL, while all the milk production traits are highly correlated?

Line 482: The author did not focus on genetic trend for 305dSNFY and 305dFY in Sahiwal cattle. Please input the results somewhere in between. Please explain why didn’t author found any steady genetic trend for all the milk production traits.

Line 496-498: The figures are not found to be numbered in the figure section!!

The authors are claiming 3 SNPs found to be involved in the milk production of Shahiwal cattle. I would like to ask whether is it possible to calculate the portion of additive genetic variance captured by the mentioned SNPs? If possible, please include their additive genetic contribution as compared to the total variance. Since, the milk production traits are complex traits with lots of genes involved and they interact with the environment to express the phenotype, therefore, if the SNPs are not significantly contributing to the phenotype, the present study would have very little impact in improving milk production traits. Therefore, these days GWAS are being performed to identify genome-wide QTLs underlying milk production traits in livestock.

Conclusion:

Line 535 and 536, please rewrite to make the sentence more readable avoiding duplication in using “is of a …….”

Reviewer #2: The manuscript is worthy and time -demanding. The experimental design, sample size, reference number and writing quality are sound enough. In introduction section, I have observed certain anomalies for nomenclature of genes. In materials and methods sections I have found certain irrelevant information for phenotypic data collection i.e. seasonal classification. The general description of the models in matrix forms incase of statistical analysis doesn't represent correctly. In figure section, it is important to mention figure legend. The following minor correction needs to be done.

Line 90: Elaborate the FecB gene

Line 92: elaborate DGAT1, GHR and ABCG2 genes

Deleted the based on the prevailing climatic condition and fodder resources available at the farms, seasons were classified as winter (December- March), summer (April-June), rainy (July-September) and autumn (October-November).

Line 179 and 180: Change y to Y.

6. PLOS authors have the option to publish the peer review history of their article (what does this mean?). If published, this will include your full peer review and any attached files.

Reviewer #1: No

Reviewer #2: **Yes: **Professor Dr. Md. Nazim Uddin, Department of Livestock Production and Management, Faculty of Veterinary, Animal and Biomedical Sciences, Sylhet Agricultural University, Sylhet-3100, Bangladesh.

---

## [Author Response · Author response to Decision Letter 0]

15 Feb 2022

Dear editors and reviewers,

We would like to express our great appreciation to you and the reviewers for the comments on our manuscript entitled "Inputs for optimizing selection platform for milk production traits of dairy Sahiwal cattle", which was submitted to PLOS ONE (Manuscript Number: PONE-D-21-30537)”. All of these comments were very helpful for revising and improving our paper. We have considered these comments carefully and have made corresponding corrections that we hope will meet with your approval. The changes in the revised manuscript are marked in blue and purple. Please see our response to the Editor and reviewer’s comments (in blue and purple).

If you have any further queries, please do not hesitate to contact us.

Kind regards,

Destaw Worku Mengistu, PhD, Assistant Professor

Animal Genetics and Breeding, 

Salale University, Salale, Ethiopia

Reply to the comments of the Editor and Reviewers for the Manuscript “Inputs for optimizing selection platform for milk production traits of dairy Sahiwal cattle” PONE-D-21-30537

Response to editor’s comments

1. Please ensure that your manuscript meets PLOS ONE's style requirements, including those for file naming. The PLOS ONE style templates can be found at https://journals.plos.org/plosone/s/file?id=wjVg/PLOSOne_formatting_sample_main_body.pdf and https://journals.plos.org/plosone/s/file?id=ba62/PLOSOne_formatting_sample_title_authors_affiliations.pdf.

Response: Respected Editor, thank you very much for providing us the link of PLOS ONE style templates. Accordingly, we have carefully revised our manuscript and edited the file naming according to the guidelines. We believe that now the manuscript is in a better condition and we hope that the revised manuscript fully adheres PLOS ONE's style requirements.

In your cover letter, please note whether your blot/gel image data are in Supporting Information or posted at a public data repository, provide the repository URL if relevant, and provide specific details as to which raw blot/gel images, if any, are not available. Email us at plosone@plos.org if you have any questions

Response: Dear Editor, after your valuable advice, we have provided the original images data for PCR amplified product of the studied genes and original uncropped and unadjusted images underlying all blot or gel results reported in the submission figure and Supporting Information files. The required results are presented in supporting information S1 Fig to S6 Fig. We further provided a separate caption for each supplementary files at the end of the manuscript revised (Line Number 674 to 704). In our cover letter, we indicated that the original image data are in Supporting information files. 

Response: Dear Editor, thanks for your question. All relevant data are within the manuscript and its supporting information files. We further uploaded the minimal underlying data set as supporting information files and the required files are presented in supporting information files (S1 File to S3 File). We have further provided a separate supporting information caption for each supplementary files at the end of the manuscript revised (Line Number 669 to 673). However, raw data of milk production and pedigree information of Sahiwal cattle used in this study cannot be shared publicly. This is because there are legal restrictions for sharing the data as the data pertains to the animal breeding experiment and not for open access sharing. This is so because it contains information regarding the selection of the animals and other important information, which is supposed to be not shared in raw form. The restrictions are imposed by the Research Advisory Council of the Institute from where the data is used for analysis (ICAR-National Dairy Research Institute, Karnal 132001 - India).

Response to reviewers’ comments

Reviewer #1

General evaluation

This is a good piece of work. A comprehensive study on exploring the genetic architecture of milk production traits in dairy cattle is of great importance in livestock industry. However, if the following issues are addressed carefully, this work could have a great impact in improving productivity and save costs in dairy industry. I suggest the author should undergo thorough revision (major) of the manuscript. 

Response: We are grateful to the reviewer for the detailed review of the manuscript and for constructive comments. The reviewer’s comments are valuable and very helpful for improving our research paper. We have carefully read all comments and have tried our best to revise the manuscript as per reviewer’s suggestions, which we hope to meet with acceptance requirements. We have gone through each and every query and carried out the major revision required, including the exact location where the change can be found in the revised manuscript. We believe that the revised version of our manuscript addresses all concerns by the reviewer in depth. The comment wise reply is given below.

Introduction:

Regarding justification of the experiment, since the author mentioned (line number 101-103) very few studies have reported the genetic association of the SNP variation in MASP2 and SIRT1 gene with milk production traits. However, it would have been a novel work, if there was no previous report of association of these two genes with milk production traits. 

Response: The authors mentioned that, very few studies have been reported on the genetic association of SNP variation in MASP2 and SIRT1 genes with milk production traits in dairy cattle. While, there was previous report of MASP2 gene with mastitis resistance and SIRT1 gene with growth traits in dairy cattle, but, genetic association of the SNP variation in MASP2 and SIRT1 genes with milk production traits in Sahiwal cattle has not been conducted yet. Respectively, the polymorphism of these specific mutation sites and its genetic associations are reported here for the first time for this breed.

Materials and method:

In line number 192, the author mentioned nine thousand and within the parenthesis mention 18000, which one is correct?

Response: Thank you for pointing this out. 

Number of iterations = 1000,000

Burn in = 100,000

One out of 50 iterations were stored for further analysis

Effective number of samples = 1000,000-100,000/50 = 18000.

Therefore, eighteen thousand is the corrected one and the text ‘nine thousand’ corrected as: Eighteen thousand. Correction have been made in the revised manuscript (Line number 185).

Results and discussion:

Line number 226: “in exon 3 region of” please replace ‘of’ with on.

Response: Needful done and “of” replaced with on (Line number 219).

Line number 233: “BstUI for MASP2 SNP at the locus” please change the line as “BstUI for MASP2 gene at the SNP locus”…………………

Response: The author thanks your kind comments. The change has been incorporated in the revised manuscript and corrected as “BstUI for MASP2 gene at the SNP locus” (Line number 225).

Line number 275: genetic diversity parameters and its corresponding Chi-squared test to determine….

Please mention the name of genetic diversity parameters such as expected heterozygosity, Observed heterozygosity, effective population size…

Response: Suggested contents have been added and incorporated in the revised section of the manuscript (Line 266-267).

Line number 288-289: “At this locus the T allele frequency was minimal” which contradicts with that given in Table:3. In table 3, T allele is shown to have allele frequency of 0.81 for T allele and 0.19 for the allele C. The allele C’s frequency would be 0.81 and T allle’s frequency would be 0.19. 

Response: The authors appreciate the reviewer comment for pointing out the error. Corrections have been done in Table 3 in the revised section of the manuscript and necessary changes indicated with blue color (Table 3). 

In line 299: The author described the genetic diversity in Sahiwal cattle as low PIC and low He indicating low genetic diversity. How will you explain the cause of low genetic diversity/ loss of genetic diversity in Sahiwal cattle? If the loss in diversity continues in this breed, this breed will face extinction. How would you solve this? Or what is your suggestion to improve the diversity? Please write in brief. 

Response: In the original manuscript, the authors mentioned that “the genetic diversity of Sahiwal cattle were low (PIC value < 0.25) to medium polymorphism level (0.25<PIC<0.50)”. 

Low genetic diversity in Sahiwal cattle may be due to the reason that not all animals mate and produce offspring, due to selective breeding in the current Sahiwal herd, where this study was conducted. Also, selection has a decreasing effect: only Sahiwal animals with a specific genetic make-up are allowed to breed. Obviously, this has consequences for the allele frequencies in the next generation, resulting in an increase in homozygosity, and thus a decrease in genetic diversity. 

To solve breed extinction due to continuous loss of genetic diversity, expansion of the size of the effective population, regular introgression from the field in to the herd (which is being practiced), control and manage relationships through controlled breeding schemes helps to maintain the genetic variation in the population. We have incorporated a new paragraph detailing how to improve the genetic in the study breed. Please see Line 290-293.

Line number 342: “Association analysis of candidate gene is a needful…..” Please rewrite or change like:

Association analysis of SNPs (variants) of candidate genes is a needful …………………….

Response: We agree with the comment and hence we have modified the sentence accordingly (Line number 306).

Line 359-360: I find no reason to mention about the statistically non-significant SNPs in the text since it has been shown in the table. Please remove part of line 359-360…

Response: Thanks for the suggestion. We have removed part of line 323-324 detailing about non-significant SNPs in the text. Moreover, we have incorporated additional text as “the SNP g.684G>T locus” at Line number 324.

Line 369: “In particular, TT genotype cows”………Please change to “In case of SIRT1 gene, TT genotype cows……

Response: We have modified the text accordingly (Line 331-332)

Line 370-371. It is suggested by the author to select heterozygotes (TC) for SNP, g-360T>C allele? How would you explain this higher performance of TC than CC and TT? Please write the explanation. 

Response: If the heterozygote TC is phenotypically different from the homozygote TT and CC because it has a higher value for a production trait or has a phenotype thought to be important for the studied breed, then the heterozygote TC can have a selective advantage due to artificial selection. When TC is specifically selected, the frequency of heterozygotes increases very quickly. However, in livestock because of strong selection favoring them as heterozygotes, some of these mutants are segregating at much higher frequencies within breeds. Overall, these mutants are generally maintained by a balance of artificial selection favouring heterozygotes and natural (and sometimes artificial) selection against mutant and/or wild-type homozygotes. However, these examples might be important for identifying other genetic variants maintained by heterozygote advantage. We have presented more explanations in the manuscript (Line 334-337) as a new paragraph.

Line number 460-462: We know, usually, the repeatability value of a trait is higher or equal to the heritability values due to the differences in the calculation formula. In repeatability calculations we account for the permanent environmental effect. Therefore, higher repeatability value obtained in the present study might be attributed to much higher impact of permanent environment in the population. Please rewrite the sentences……….

Response: We agree with the comment and hence we have modified the sentences accordingly (Line number 424-427)

Line number 482, Please rewrite the sentence as: in general, EBVs for 305dmY, LMY and LL were increased………………

Response: We have now modified the sentence accordingly (Line number 446). 

Line 485-486: The meagre change in response is due to low-selection intensity………………is in contrast to the line 280-282, where the author explained the reason for loss of genetic diversity due to increased selection pressure. Please rewrite the sentences and explain carefully. 

Response: Thanks for the suggestion. Suggested contents have been explained and incorporated in the revised section of the manuscript (Line 449-452). 

Line 482: would you please explain the reason for very low genetic progress in LL, while all the milk production traits are highly correlated?

Response: In this study, we observed the existence of strong genetic correlation between milk production traits, implying that these traits can be improved genetically through indirect selection when selecting for any of the traits. Low genetic progress over time for LL trait imply that, there seems to be other factors affecting this trait, for which the influence of managerial and temporal environmental factors is more evident. Standard LL beyond 305 days is not desired as the milk yield from the animals who extended their lactation, goes on decreasing. Thus, the increment in LL was not a preferred over a certain limit. In a managed herd, a common practice of drying off the animals is also practiced, so that the animal should be in regular reproduction cycle, this also discourages extension of the LL.

Line 482: The author did not focus on genetic trend for 305dSNFY and 305dFY in Sahiwal cattle. Please input the results somewhere in between. Please explain why didn’t author found any steady genetic trend for all the milk production traits.

Response: In India current selection objectives emphasized mainly on milk yield, with enough selection pressure on fat percentage to maintain composition of milk at legal standards. Since 305-days milk yield, 305-days fat yield and 305-days solid not fat yield traits are genetically fairly closely related (that is, their genetic correlations are high and positive), similar magnitude of genetic trend is expected for 305dFY and 305dSNFY in Sahiwal cattle, even though these were ignored in the selection program. For instance, inclusion of genetic trend for these traits receives little attention to assess the progress made due to selection over the years. As a result, we respectfully did not include information on the genetic trend for 305dSNFY and 305dFY in Sahiwal cattle. 

Even though the genetic trend per year is not steady, the genetic progress per generation (not presented in the manuscript) is increased in Sahiwal cattle. This imply that there is a balance between increasing accuracy of selection and the time required to achieve the information to achieve the largest genetic gain per year. Examples of generation wise genetic trend for 305-dMY is attached here below.

Line 496-498: The figures are not found to be numbered in the figure section!!

Response: Needful done in the figure section of the revised manuscript

The authors are claiming 3 SNPs found to be involved in the milk production of Shahiwal cattle. I would like to ask whether is it possible to calculate the portion of additive genetic variance captured by the mentioned SNPs? If possible, please include their additive genetic contribution as compared to the total variance. Since, the milk production traits are complex traits with lots of genes involved and they interact with the environment to express the phenotype, therefore, if the SNPs are not significantly contributing to the phenotype, the present study would have very little impact in improving milk production traits. Therefore, these days GWAS are being performed to identify genome-wide QTLs underlying milk production traits in livestock.

Response: While we appreciate the reviewer’s feedback and query about the additive variance of SNPs, but for this analysis, we respectfully point out that, we have genotyped 150 lactating Sahiwal animals only. Estimating the portion of additive genetic variance of SNPs with small number of genotyped animals will not give good results, leading to wrong estimation of additive genetic variance explained by the mentioned SNPs as compared to total variance. As a result (co)variance analysis with SNP as a direct additive effect was not considered and explained for this analysis. Even though, GWAS are being performed to identify genome-wide QTLs underlying milk production traits in livestock, the authors tried to explain (Introduction section) the importance of a single gene effect in the identification of limited number of causal genes. Research on candidate gene approach increases the number of genes with a known major effect in dairy cattle. For example, beta-lacto globulin alleles have a marked effect on the efficiency of cheese production. The Diacylglycerol acyl-transferase 1 (DGAT1) alleles influence fat percentage in milk and the milk fat composition. Therefore, those 3 SNPs involved in the milk production traits of Sahiwal cattle would have positive impact for improving milk production traits in dairy cattle. In conclusion section of the manuscript, authors strongly recommend to validate those SNPs on large sample size population to be included in the selection program.

Conclusion: 

Line 535 and 536, please rewrite to make the sentence more readable avoiding duplication in using “is of a …….”

Response: We have revised and re-written the conclusion to address your concerns. Many improvements have been done. Please see the conclusion section in the revised manuscript (Line 498 and 499).

Reviewer #2

The manuscript is worthy and time demanding. The experimental design, sample size, reference number and writing quality are sound enough. 

Response: Authors are thankful for the comment.

In introduction section, I have observed certain anomalies for nomenclature of genes. In materials and methods sections I have found certain irrelevant information for phenotypic data collection i.e. seasonal classification. The general description of the models in matrix forms incase of statistical analysis doesn’t represent correctly. In figure section, it is important to mention figure legend.

Response: Dear reviewer, thank you very much for your insightful suggestions. We have carefully read all comments and have tried our best to revise the manuscript as per reviewer suggestions, which we hope to meet with acceptance requirements. As suggested by the reviewer, the authors have done the necessary modifications in the introduction, material and methods and results and discussion section of the revised manuscript. Moreover, figure legends are incorporated in the figure section of revised manuscript. All changes are highlighted in purple in the revised manuscript. 

The following minor corrections needs to be done. 

Line 90: Elaborate the FecB gene

Response: FecB gene has been elaborated in the revised manuscript (Line 90)

Line 92: Elaborate DGAT1, GHR and ABCG2 genes 

Response: DGAT1, GHR and ABCG2 genes have been elaborated in the revised manuscript (Line 92-93).

Deleted the based on the prevailing climatic condition and fodder resources available at the farms, seasons were classified as winter (December-March), summer (April-June), rainy (July-September) and autumn (October- November).

Response: We have deleted the paragraph detailing season classifications.

Line 179 and 180: Change y to Y.

 Response: y changed to Y in Line 178 and 179

All the above changes have been reflected in the manuscript in blue and purple text.

---

## [Decision Letter · Decision Letter 1]

18 Apr 2022

Inputs for optimizing selection platform for milk production traits of dairy Sahiwal cattle

PONE-D-21-30537R1

Dear Dr. Destaw Worku,

We’re pleased to inform you that your manuscript has been judged scientifically suitable for publication and will be formally accepted for publication once it meets all outstanding technical requirements.

Kind regards,

Md Ashrafuzzaman, Ph.D.

Academic Editor

PLOS ONE

---

## [Editor Report · Acceptance letter]

12 May 2022

PONE-D-21-30537R1 

Inputs for optimizing selection platform for milk production traits of dairy Sahiwal cattle 

Dear Dr. Worku:

I'm pleased to inform you that your manuscript has been deemed suitable for publication in PLOS ONE. Congratulations! Your manuscript is now with our production department. 

Kind regards, 

on behalf of

Dr. Md Ashrafuzzaman 

Academic Editor

PLOS ONE